# Prognostic Value of Pathologically Positive Nodal Number in p16-Negative Oropharyngeal and Hypopharyngeal Squamous Cell Carcinoma with pN3b Status

**DOI:** 10.3390/diagnostics12061443

**Published:** 2022-06-12

**Authors:** Ming-Hsien Tsai, Yu-Tsai Lin, Hui-Ching Chuang, Tai-Ling Huang, Hui Lu, Chih-Yen Chien, Fu-Min Fang

**Affiliations:** 1Department of Otolaryngology, Kaohsiung Chang Gung Memorial Hospital and Chang Gung University College of Medicine, Kaohsiung 83301, Taiwan; b9302094@cgmh.org.tw (M.-H.T.); xeye@cgmh.org.tw (Y.-T.L.); entjulia@cgmh.org.tw (H.-C.C.); luhui88@cgmh.org.tw (H.L.); 2Kaohsiung Chang Gung Head and Neck Oncology Group, Cancer Center, Kaohsiung Chang Gung Memorial Hospital, Kaohsiung 83301, Taiwan; victor99@cgmh.org.tw; 3College of Pharmacy and Health Care, Tajen University, Pingtung County 90741, Taiwan; 4Department of Medicine, Chang Gung University College of Medicine, Taoyuan 33302, Taiwan; 5Department of Hematology and Oncology, Kaohsiung Chang Gung Memorial Hospital and Chang Gung University College of Medicine, Kaohsiung 83301, Taiwan; 6Institute for Translational Research in Biomedicine, Kaohsiung Chang Gung Memorial Hospital, Kaohsiung 83301, Taiwan; 7Department of Radiation Oncology, Kaohsiung Chang Gung Memorial Hospital and Chang Gung University College of Medicine, Kaohsiung 83301, Taiwan

**Keywords:** pathologically positive nodal number, extranodal extension, oropharyngeal cancer, hypopharyngeal cancer, nomogram

## Abstract

In this study, we aimed to investigate the prognostic value of the number of pathologically positive nodes (pN+) in p16-negative oropharyngeal (OPSCC) and hypopharyngeal (HPSCC) squamous cell carcinoma cases with pN3b status after surgery. We reviewed the clinical and pathological features of 120 newly diagnosed p16-negative OPSCC and HPSCC patients with pN3b status after radical surgery. The primary endpoints were the 5-year overall survival (OS), cancer-specific survival (CSS), and their prognostic factors. We used the Cox proportional hazards model for survival analysis. We generated predictive nomograms that incorporated the clinicopathological factors of OS and CSS. The 5-year OS and CSS rates were 44.1% and 59.1%, respectively. The optimal number of pN+ to predict the 5-year OS and CSS was pN+ = 3. In the Cox model, we observed that pN+ ≥ 3 was a significantly negative predictor of OS (HR: 1.9, 95% CI: 1.1–3.2, *p* = 0.021) and CSS (HR: 2.3; 95% CI: 1.2–4.6; *p* = 0.015). After adding the pN+ variable, the c-index of the predictive nomogram incorporating assorted clinicopathological factors increased from 0.66 to 0.689 for OS and from 0.713 to 0.75 for CSS. The results highlight the prognostic value of the pN+ number in p16-negative OPSCC and HPSCC patients with pN3b status.

## 1. Introduction

Pathologically positive nodal metastasis is a well-known poor prognostic factor in patients with head and neck squamous cell cancer (HNSCC) [1]. The presence of extranodal extension (ENE) is an adverse factor in patients with nodal metastases, causing a poor rate of regional control and an increasing rate of distant metastases [2]. Both human papillomavirus (HPV)-negative oropharyngeal squamous cell carcinoma (OPSCC) and hypopharyngeal squamous cell carcinoma (HPSCC) are aggressive types of HNSCC. The American Joint Committee on Cancer (AJCC) staging system 8th edition introduced remarkable changes to nodal classification regarding the presence of ENE for patients with HPV-negative OPSCC and HPSCC [3], which upgrade pN1 with ENE to pN2a and pN2 with ENE to pN3b, respectively. However, this nodal classification may underestimate the effect of the metastatic nodal number on pN3b status. According to the AJCC 8th edition classification, HNSCC patients with two nodal metastases are staged in the same way as those with twenty nodal metastases; however, the biological behavior and survival prognosis might be quite different between the two groups. The larger the number of metastatic lymph nodes in HNSCC patients, the more inferior the survival outcomes [4,5,6,7]. However, most of these studies enrolled heterogeneous cases with positive nodal cases, without specific investigations into the subgroup of cases with pN3b status. In this study, we investigated the prognostic value of the metastatic nodal number in a cohort of p16-negative OPSCC and HPSCC patients with pN3b status, and we attempted to generate prognostic nomograms that incorporated the quantitative metastatic nodes and other clinicopathological features to estimate the 5-year overall survival (OS) and cancer-specific survival (CSS).

## 2. Materials and Methods

### 2.1. Study Design

According to the cancer database of the institute, there were 340 previously untreated p16-negative OPSCC and HPSCC patients who received radical surgery with unilateral/bilateral neck dissection from January 2007 to March 2016 at Kaohsiung Chang Gung Memorial Hospital, Taiwan. We excluded patients who had a distant metastasis, a previous history of other cancer, prior treatment for HNSCC, received inadequate neck dissection, or received neoadjuvant chemotherapy and/or radiotherapy before radical surgery. The p16 expression in tumor cells was determined by immunohistochemistry, according to the criteria of the AJCC 8th edition. A total of 237 patients had pathologically nodal metastases: 41 (17.3%) cases were staged as pN1, 76 (32.1%) as pN2, and 120 (50.6%) as pN3b. We retrospectively reviewed the detailed information on the clinical and pathological features of the 120 patients with pN3b status.

### 2.2. Variables and Endpoints

The clinical and pathological variables collected included sex, age, lifestyle factors (habits of alcohol drinking, betel nut chewing, and smoking cigarettes), primary cancer location, pT classification, tumor differentiation, surgical margin, perineural invasion (PNI), lymphovascular invasion (LVI), lymph node yields, maximal size of pN+, and the number of pN+. The primary endpoints were the 5-year OS, CSS, and their prognostic factors.

### 2.3. Statistical Analysis

We used the Kaplan–Meier method to estimate the probability of survival, and the log-rank test to examine the statistical significance between the groups. We used the Cox proportional hazards model for multivariate analysis, and calculated the hazard ratios (HRs) and 95% confidence intervals (CIs) for each predictor. We considered two-tailed test with a *p*-value < 0.05 as statistically significant. We performed statistical processing using SPSS 25.0 software (SPSS/IBM, Inc., Chicago, IL, USA).

To generate predictive nomograms of OS and CSS, we incorporated variables including cancer location, pT classification, PNI, LVI, surgical margin, adjuvant therapy or not, and the number of metastatic nodes into the model by using the R software “rms” package (Version 5.1–0, Vanderbilt University, Nashville, TN, USA). To validate the nomograms, we calculated the concordance index (c-index) for the conventional TNM staging, as well as for the proposed nomogram models with and without the number of positive nodes, to assess the accuracy of the nomogram in predicting OS and CSS, where values of 0.5 and 1.0 signified random and perfect predictability, respectively. We created calibration plots to determine whether the predicted survival was consistent with the actual observed survival.

## 3. Results

### 3.1. Patient Characteristics

The clinical and pathological characteristics of the study patients are summarized in Table 1. The median age was 53 years (range: 35–78 years), with 116 (96.7%) men and 4 (3.3%) women patients. Most of the patients had a habit of smoking (*n* = 113, 94.2%), betel nut chewing (*n* = 103, 85.8%), and/or alcohol drinking (*n* = 110, 91.7%). The primary cancer locations were the hypopharynx in 64 (53.3%) patients and the oropharynx in 56 (46.7%) patients. Most of the cases were moderately differentiated SCC (*n* = 89, 74.2%), followed by well-differentiated SCC (*n* = 24, 20%), and poorly differentiated SCC (*n* = 7, 5.8%). With regard to the distribution of the pT classification, we found 17 (14.2%) pT1, 35 (29.1%) pT2, 29 (24.2%) pT3, and 39 (32.5%) pT4 cases. PNI and LVI were reported in 62 (51.7%) and 73 (62.8%) cases, respectively. In total, 67 (55.8%) patients experienced ipsilateral neck dissection, and 53 (44.2%) patients underwent bilateral neck dissection. The median number of metastatic nodes was 3 (range: 1–41). The median number of lymph node yields from neck dissection was 32 (range: 8–98), and 105 (87.5%) patients had lymph node yields ≥ 18. Thirty-seven (30.8%) cases had bilateral nodal metastasis. The median size of the largest metastatic node in each case was 2.8 cm (range: 0.6–7 cm). Postoperative adjuvant therapy was performed in 108 cases, including 98 patients (81.7%) by concurrent chemoradiotherapy (CCRT), and 10 patients (8.3%) by radiotherapy alone. The remaining 12 patients (10%) did not receive any adjuvant therapy due to high comorbidity or the refusal of the patients. The median follow-up period was 42.3 months (range: 0.4–147.1 months).

### 3.2. Number of pN+ and Survival Outcome

During the follow-up period, treatment failure was observed in 45 (37.5%) patients. The patterns of treatment failure included local alone (*n* = 5), regional alone (*n* = 13), distant alone (*n* = 15), locoregional (*n* = 7), regional and distant (*n* = 2), locoregional and distant (*n* = 2), and local and distant (*n* = 1). The median times the regional and distant failures were 8.2 months (range: 3.2–58.4 months) and 14.6 months (range: 4.5–53.2 months), respectively. A total of 40 patients died from the disease, and 6 died from treatment-related complications during the follow-up period. The 5-year OS and CSS rates were 44.1% and 59.1%, respectively. The optimal cut-off number of pN+ to predict the 5-year OS and CSS rates was pN+ = 3 (OS: *p* = 0.042; CSS: *p* = 0.045; Figure 1). Patients with pN+ ≥3 were more frequently those with pT3–4, LVI, or bilateral nodal disease (Table 2). The subgroup analysis of OS and CSS is presented in Table 3. For those with pN+ < 3, we observed that the 5-year OS and CSS rates were statistically significantly superior to those with pN+ ≥ 3 (Figure 2; OS: 57.9% versus 35.9%, *p* = 0.012; CSS: 75.7% versus 49.0%, *p* = 0.013). The variables of no adjuvant therapy and advanced pT classification were unfavorable predictors of both OS and CSS (*p* < 0.05), and a close surgical margin (less than 5 mm) was an unfavorable predictor of OS (*p* < 0.05). In the Cox model (Table 4), we observed that pN+ ≥ 3 was the only statistically significant unfavorable predictor of OS (HR: 1.9; 95% CI: 1.1–3.2; *p* = 0.021) and CSS (HR: 2.3; 95% CI: 1.2–4.6; *p* = 0.015). Adjuvant therapy was also a favorable predictor of CSS (HR: 0.4; 95% CI: 0.2–0.9; *p* = 0.048), and a surgical margin < 5 mm was an unfavorable predictor of OS (HR: 1.8; 95% CI: 1.1–3.0; *p* = 0.017).

### 3.3. Predictive Nomograms of Survival

Figure 3A,C depict the predictive nomograms for the 5-year OS and CSS, incorporating the following variables: number of pN+ (≥3 or <3), pT classification (pT1, pT2, pT3, or pT4), PNI (presence or absence), LVI (presence or absence), surgical margin (<5 mm or ≥5 mm), cancer location (hypopharynx or oropharynx), and adjuvant therapy (yes or no). In this scenario, if a case of hypopharyngeal cancer had pN+ ≥ 3, pT4, the presence of PNI, the presence of LVI, surgical margin < 5 mm, and occurred after adjuvant therapy, the predicted 5-year OS and CSS rates would be 32.9% and 46.2%, respectively. The calibration curves show that the 5-year OS and CSS predicted by the nomogram were consistent with actual observations (Figure 3B,D). The nomogram model for the OS based only on the AJCC stage had a c-index of 0.582 (95% CI: 0.515−0.649). If the model incorporated assorted clinicopathological factors without pN+, the c-index was 0.66 (95% CI: 0.597−0.723), which would increase to 0.689 (95% CI: 0.626−0.752) if pN+ were included. For CSS, the nomogram model based only on the AJCC stage had a c-index of 0.610 (95% CI: 0.530−0.690). If the model incorporated assorted clinicopathological factors without pN+, the c-index was 0.713 (95% CI: 0.639−0.787), which would increase to 0.75 (95% CI: 0.681−0.819) if pN+ was included.

## 4. Discussion

ENE is a well-known independent risk factor of locoregional failure and survival outcomes in HPV-negative OPSCC and HPSCC patients [8,9]. p16-negative OPSCC and HPSCC are classified in the same category in the AJCC staging system 8th edition, in which the subgroup with pN3b status includes heterogenous nodal groups with ENE, but without further stratification on the basis of the number of pN+. The survival predictability based on the stratification of AJCC staging system 8th edition has been challenged in several reports for patients with oral squamous cell carcinoma (OSCC) [10,11] or laryngohypopharyngeal cancers [12] after radical surgery. When accounting for the number of pN+, we observed that the classic prognostic factors of the nodal status that are used by the AJCC staging (i.e., nodal size and contralaterality) were no longer independent predictors of survival. Our findings echo the finding that the number of pN+ is a critical predictor of OS and CSS for HPV-negative OPSCC and HPSCC patients after adjustment for the other clinicopathological features in the Cox model or predictive nomogram.

We observed a statistically significant trend in most of the studies, where the HNSCC patients with more pN+ presented poorer survival outcomes; however, the optimal cutoff number of pN+ varied in the different study components. Hua et al. studied 81 patients with surgically treated hypopharyngeal cancer, and their subgroup analysis revealed that those with pN+ ≥ 4 had inferior overall survival in 64 pathologically proven nodal-positive patients [6]. Roberts et al. studied surgically treated OPSCC patients between 2004 and 2012 in the Surveillance, Epidemiology, and End Results (SEER) database and they showed that pN+ > 5 had poorer OS than the control group [4]. Ho et al. examined the data of 1332 HPSCC patients with pN+ from the National Cancer Database (NCDB), and, also, observed that patients with pN+ ≤ 5 had better OS than those with pN+ > 5 [12]. Furthermore, Liao et al. reported on a cohort of 365 OSCC patients with pN3b disease, and they observed that those patients with pN+ ≥ 8 had an unfavorable prognosis compared with their counterparts [11]. By contrast, in our study, which focused on the subgroup of p16-negative OPSCC and HPSCC patients with pN3b status, we observed that the optimal cutoff number to predict survival was pN+ = 3.

The lymph node yield from neck dissection has also been associated with the survival outcome for HNSCC patients [13,14]. A lymph node yield ≥ 18 was significantly associated with both decreased recurrence and improved survival [15,16]. However, we did not observe this variable to be a significant predictor of OS and CSS in our cohort, which could be explained by the majority (87.5%) of our patients having lymph node harvests ≥ 18.

A positive/close surgical margin is an important survival prognosticator for patients with HNSCC. Jacques Bernier et al. revealed that a microscopically involved surgical margin is a negative independent prognostic factor in HNSCC patients [17]. Eldeeb et al. studied a series of 413 patients with HNSCC who underwent surgical resection, and the patients with positive/close surgical margins had poor recurrence-free survival and OS, regardless of the other tumor or patient characteristics [18]. In our previous study, we conducted a retrospective analysis of 90 patients who had undergone salvage total laryngectomies, and we showed that the patients with tumor-free margins had better OS and CSS [19]. In the study on the subgroup of patients with p16-negative OPSCC and HPSCC patients with pN3b disease, we observed that the surgical margin remained a significant survival predictor.

The nomogram, which is a dependable predictive tool, is extensively employed in oncology research, including in HNSCC [20]. The nomogram integrates the demographic and pathological information with better patient stratification to estimate individualized treatment outcomes. To the best of our knowledge, few studies in our literature review incorporated the pN+ variable into the predictive nomogram for HNSCC patients. According to our proposed nomogram, the clinicians can execute individualized estimations of the risk of overall and cancer-specific mortality for this group of patients and provide personalized treatment strategies.

This study had several limitations. First, the cases were limited to a single institute, and the findings are therefore vulnerable to selection bias. Second, the exact number of ENE-positive nodes was lacking in our cohort due to its retrospective design. However, our data provide evidence that the pN+ number is a critical survival predictor for p16-negative OPSCC and HPSCC patients with N3b status, and they highlight the prognostic value of the pN+ number when the conventional AJCC staging system is applied for survival prediction.

## 5. Conclusions

Our data show that the pN+ number is a critical survival predictor for p16-negative OPSCC and HPSCC patients with N3b status. Stratification on the basis of the number of pN+ should be considered when the conventional AJCC staging system is applied for survival prediction.

## Figures and Tables

**Figure 1 diagnostics-12-01443-f001:**
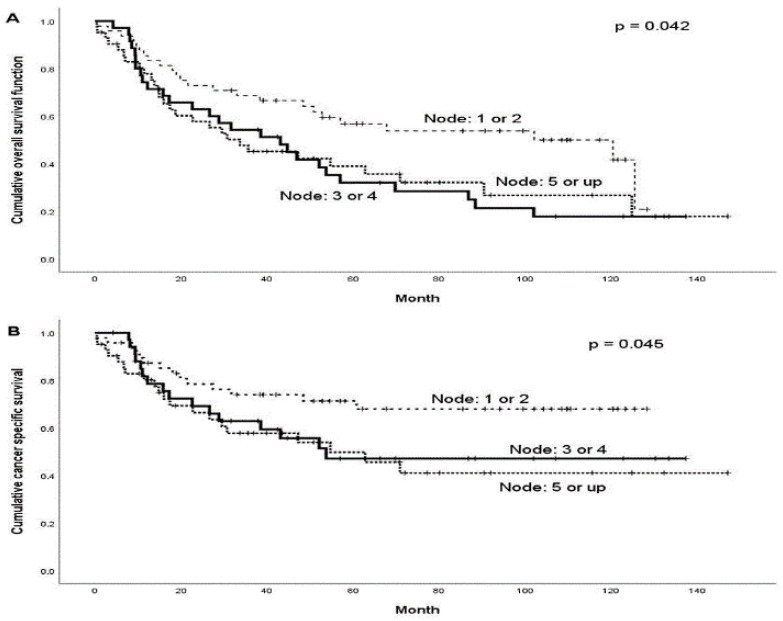
Kaplan–Meier survival curves according to different numbers of positive nodes: (**A**) overall survival curves and (**B**) cancer-specific survival curves.

**Figure 2 diagnostics-12-01443-f002:**
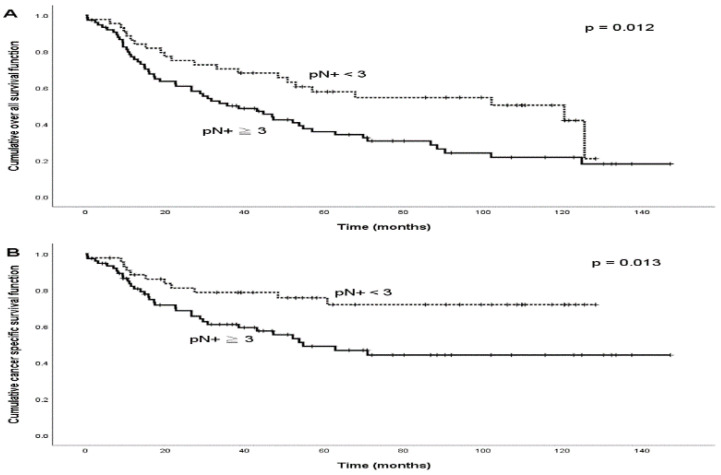
Kaplan–Meier survival curves among p16-negative OPSCC and HPSCC patients with two different categories of numbers of positive nodes (pN+): (**A**) overall survival curves and (**B**) cancer-specific survival curves.

**Figure 3 diagnostics-12-01443-f003:**
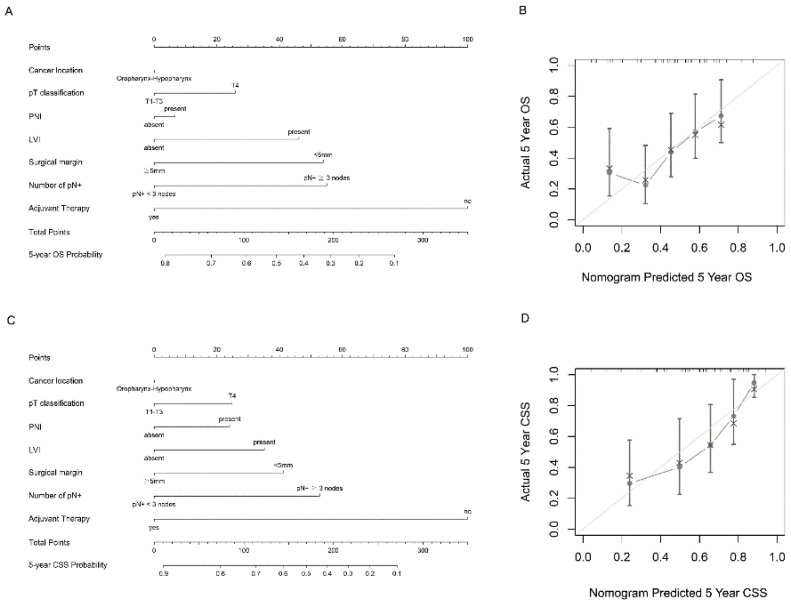
Nomogram and survival predictions: (**A**) nomogram for OS prediction and (**C**) nomogram for CSS prediction. A vertical line is drawn from each factor to the point score. By adding the points from all factors, a total points score is reached, which is translated into 5-year OS probabilities by drawing a vertical line to its axis. (**B**) Calibration plots of the nomogram to predict 5-year OS, and (**D**) calibration plots of the nomogram to predict 5-year CSS. The 45-degree straight line indicates the ideal prediction, and the dark-gray line represents the nomogram’s performance. Gray dots with bars represent the nomogram’s performance with 95% CI when applied to the observed surviving cohorts.

**Table 1 diagnostics-12-01443-t001:** Patient characteristics (*n* = 120).

Characteristic	Value	%
Age, median range (years)	53 (35–78)	
Sex (male/female)	116/4	96.7/3.3
Smoking habit (yes)	113	94.2
Betel nut chewing habit (yes)	103	85.8
Alcohol drinking habit (yes)	110	91.7
Cancer location (oropharynx/hypopharynx)	56/64	46.7/53.3
pT classification (T1–2/T3–4)	52/68	43.3/56.7
Histologic grade (WDSCC/MDSCC/PDSCC)	24/89/7	20.0/74.2/5.8
Perineural invasion (yes)	62	51.7
Lymphovascular invasion (yes)	73	60.8
Lymph node yields ≥ 18	105	87.5
Maximal size of pN+ (≤3 cm/3–6 cm/>6 cm)	70/47/3	58.3/39.2/2.5
Margin (≥5 mm/<5 mm)	75/45	62.5/37.5
Neck dissection (ipsilateral/bilateral)	67/53	55.8/44.2
Bilateral nodal disease (yes)	37	30.8
Adjuvant therapy (CCRT/RT alone/none)	98/10/12	81.7/8.3/10.0

WDSCC: well-differentiated squamous cell carcinoma; MDSCC: moderately differentiated squamous cell carcinoma; PDSCC: poorly differentiated squamous cell carcinoma; CCRT: concurrent chemoradiotherapy; RT: radiotherapy.

**Table 2 diagnostics-12-01443-t002:** Association between clinicopathological factors and number of pN+.

Variable	pN+ < 3	pN+ ≥ 33	*p* Value
Sex (male/female)	42/2	74/2	0.623
Age (<53 years/≥53 years)	22/22	37/39	0.889
Cancer location (oropharynx/hypopharynx)	23/21	33/43	0.349
pT classification (T1–2/T3–4)	27/17	25/51	0.002
Histologic grade (WDSCC/MDSCC/PDSCC)	10/31/3	14/58/4	0.742
Perineural invasion (no/yes)	26/18	32/44	0.117
Lymphovascular invasion (no/yes)	24/20	23/53	0.012
Margin (≥5 mm/<5 mm)	15/29	30/46	0.557
Bilateral nodal disease (no/yes)	38/6	45/31	0.002
Lymph node yields (<18/≥18)	7/37	8/68	0.39
Maximal size of pN+ (≤3 cm/3–6 cm/>6 cm)	20/23/1	50/24/2	0.061
Adjuvant therapy (no/yes)	5/39	7/69	0.757

WDSCC: well-differentiated squamous cell carcinoma; MDSCC: moderately differentiated squamous cell carcinoma; PDSCC: poorly differentiated squamous cell carcinoma; pN+: pathologically positive nodes.

**Table 3 diagnostics-12-01443-t003:** Univariate survival analysis for p16-negative OPSCC and HPSCC patients with pN3b status.

Variable	5-Year OS (%)	*p*-Value	5-Year CSS (%)	*p*-Value
Sex (male/female)	43.8/50.0	0.560	59.6/50.0	0.721
Age (<53 years/≥53 years)	46.5/41.5	0.853	55.6/62.3	0.468
Cancer location (oropharynx/hypopharynx)	37.5/49.7	0.492	51.9/65.6	0.144
pT classification (T1–2/T3–4)	53.2/37.3	0.046	71.6/50.1	0.041
Histologic grade (WDSCC/MDSCC/PDSCC)	40.9/43.3/71.4	0.944	57.2/59.1/71.4	0.950
Perineural invasion (no/yes)	48.2/38.3	0.399	65.8/51.2	0.121
Lymphovascular invasion (no/yes)	51.3/38.4	0.093	67.6/52.7	0.199
Margin (≥5 mm/<5 mm)	48.0/41.8	0.035	63.5/56.8	0.09
Bilateral nodal disease (no/yes)	47.0/37.7	0.118	61.5/55.2	0.298
Lymph node yields (<18/≥18)	33.3/45.8	0.223	54.5/60.2	0.933
Maximal size of pN+ (≤3 cm/3–6 cm/>6 cm)	39.1/50.7/66.7	0.220	54.8/65.1/66.7	0.447
Number of pN+ (<3/≥3)	57.9/35.9	0.012	75.7/49.0	0.013
Adjuvant therapy (no/yes)	41.7/44.2	0.031	50.0/60.3	0.047

OS: overall survival; CSS: cancer-specific survival; WDSCC: well-differentiated squamous cell carcinoma; MDSCC: moderately differentiated squamous cell carcinoma; PDSCC: poorly differentiated squamous cell carcinoma; pN+: pathologically positive nodes.

**Table 4 diagnostics-12-01443-t004:** Multivariate survival analysis for p16-negative OPSCC and HPSCC patients with pN3b status.

	5-Year OS		5-Year CSS	
	HR (95% CI)	*p*	HR (95% CI)	*p*
Variable				
pT classification: T3–4 (ref: T1–2)	1.4 (0.9–2.4)	0.152	N/A	N/A
Surgical margin: <5 mm (ref: ≥5 mm)	1.8(1.1–3.0)	0.017	N/A	N/A
pN+: ≥3 (ref: <3)	1.9 (1.1–3.2)	0.021	2.3 (1.2–4.6)	0.015
Adjuvant therapy: yes (ref: no)	0.6 (0.3–1.1)	0.088	0.4 (0.2–0.9)	0.048

OPSCC: oropharyngeal squamous cell carcinoma; HPSCC: hypopharyngeal squamous cell carcinoma; OS: overall survival; CSS: cancer-specific survival; pN+: pathologically positive nodes.

## Data Availability

The data presented in this study are available upon request from the corresponding author. The data are not publicly available due to ethical restrictions.

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
