# Peer review of "Prognostic Value of Pathologically Positive Nodal Number in p16-Negative Oropharyngeal and Hypopharyngeal Squamous Cell Carcinoma with pN3b Status"

_diagnostics, 2022, doi:10.3390/diagnostics12061443_

Round 1
Reviewer 1 Report
Summary
This paper attempted to investigate the prognostic value of number of pathologically positive node in p-16 negative oropharyngeal and hypopharyngeal squamous cell carcinoma patients with pN3b status after surgery.
General concept comments
In terms of introduction, the study was carefully designed. It highlighted the limitation of the new AJCC N staging system, especially the importance of ENE.
In terms of methods, the inclusion and exclusion criteria are scientifically sound. The number of patients recruited are solid.
In terms of results, the information and results have been presented in various ways, including figures, tables, nomograms, KM survival curves, etc.
Overall, the English needs improvement, especially grammar and vocabulary.
Specific comments
Page 2, line 1, ‘type’ should be types.
Page 2, 2.1 Study design. Was the status of distant metastasis (M) included or excluded in the criteria? Surely some pN3b patients would be M+.
Page 3, ‘pT classification, there were 17 pT1, 35 pT2, 29 pT3 and 39 pT4, respectively’. Please also quote the percentage of each pT classification.
Page 8, ‘Our data show that pN+ is a critical survival predictor of for p-16 negative OPSCC’. Delete ‘of’.
Author Response
English language and style
Comment : (x) Moderate English changes required
Response: Thanks for the suggestion. The English editing was done.
General concept comments
Comment: In terms of introduction, the study was carefully designed. It highlighted the limitation of the new AJCC N staging system, especially the importance of ENE. In terms of methods, the inclusion and exclusion criteria are scientifically sound. The number of patients recruited are solid. In terms of results, the information and results have been presented in various ways, including figures, tables, nomograms, KM survival curves, etc. Overall, the English needs improvement, especially grammar and vocabulary.
Response: Thanks for the suggestion. The English editing was done.
Specific comments
Specific comment 1: Page 2, line 1, ‘type’ should be types.
Response: Thanks for the kindly editing. The grammar was corrected. (Page 1, line 45).
Specific comment 2: Page 2, 2.1 Study design. Was the status of distant metastasis (M) included or excluded in the criteria? Surely some pN3b patients would be M+.
Response: We thank for this insightful observation. Those with distant metastasis were excluded. We added it in the page 2, line 22.
Specific comment 3: Page 3, ‘pT classification, there were 17 pT1, 35 pT2, 29 pT3 and 39 pT4, respectively’. Please also quote the percentage of each pT classification.
Response: We thank the reviewer for this suggestion. We corrected it in the page 3, line 13-14.
Specific comment 4: Page 8, ‘Our data show that pN+ is a critical survival predictor of for p-16 negative OPSCC’. Delete ‘of’.
Response: Thanks for kindly editing. We corrected it.
Reviewer 2 Report
The paper is well done. The authors show how massive lymph node involvement has a dramatic impact on the prognosis of these patients.Author Response
English language and style
Comment: (x) Extensive editing of English language and style required
Response: Thanks for the suggestion. The English editing was done.
Comments and Suggestions for Authors
Comment: The paper is well done. The authors show how massive lymph node involvement has a dramatic impact on the prognosis of these patients
Response: Thanks.